# Bayesian Quantile and Expectile Optimisation

**Victor Picheny**[1]        **Henry Moss**[1]        **Léonard Torossian**[2]        **Nicolas Durrande**[1]

[1]Secondmind Labs, Cambridge, UK
[2]Inria, Université Côte d'Azur, France

## Abstract

Bayesian optimisation (BO) is widely used to optimise stochastic black box functions. While most BO approaches focus on optimising conditional expectations, many applications require risk-averse strategies and alternative criteria accounting for the distribution tails need to be considered. In this paper, we propose new variational models for Bayesian quantile and expectile regression that are well-suited for heteroscedastic noise settings. Our models consist of two latent Gaussian processes accounting respectively for the conditional quantile (or expectile) and the scale parameter of an asymmetric likelihood functions. Furthermore, we propose two BO strategies based on max-value entropy search and Thompson sampling, that are tailored to such models and that can accommodate large batches of points. Contrary to existing BO approaches for risk-averse optimisation, our strategies can directly optimise for the quantile and expectile, without requiring replicating observations or assuming a parametric form for the noise. As illustrated in the experimental section, the proposed approach clearly outperforms the state of the art in the heteroscedastic, non-Gaussian case.

## 1 INTRODUCTION

Let $\Psi : \mathcal{X} \times \Omega \to \mathbb{R}$ be an unknown function, where $\mathcal{X} \subset [0,1]^D$ and $\Omega$ denotes a probability space representing some uncontrolled variables. For any fixed $x \in \mathcal{X}$, $Y_x = \Psi(x, \cdot)$ is a random variable of distribution $\mathbb{P}_x$. We assume here a classical *black-box optimisation* framework: $\Psi$ is available only through (costly) pointwise evaluations of $Y_x$. Typical examples may include stochastic simulators in physics or biology (see Skullerud [1968] for simulations of ion motion and Székely Jr and Burrage

[2014] for simulations of heterogeneous natural systems), but $\Psi$ can also represent the performance of a machine learning algorithm according to some hyperparameters (see [see Bergstra et al., 2011, for instance]. In the latter case, the randomness can come from the use of minibatching in the training procedure, the choice of a stochastic optimiser or the randomness in the initialisation of the optimiser.

Let $g(x) = \rho(\mathbb{P}_x)$ be the objective function we want to maximise, where $\rho$ is a real-valued functional defined on probability measures. The canonical choice for $\rho$ is the expectation, which is sensible when the exposition to extreme values is not a significant aspect of the decision. However, in a large variety of fields such as agronomy, medicine or finance, decision makers have an incentive to protect themselves against extreme events since they may lead to severe consequences. To take these rare events into account, one should consider alternative choices for $\rho$ that can capture the behaviour of the tails of $\mathbb{P}_x$, such as the quantile [Rostek, 2010], conditional value-at-risk (CVaR, see Rockafellar et al. [2000]) or expectile [Bellini and Di Bernardino, 2017]. In this paper we focus our interest on the modelling and optimisation of quantiles and expectiles.

Given an estimate of $g$ based on available data, global optimisation algorithms define a policy that finds a trade-off between exploration and intensification. More precisely, the algorithm has to explore the input space in order to avoid getting trapped in a local optimum, but it also has to concentrate its budget on input regions identified as having a high potential. The latter results in accurate estimates of $g$ in the region of interest and allows the algorithm to return an optimal input value with high precision.

In the context of Bayesian optimisation (BO), such trade-offs have been initially studied by Mockus et al. [1978] and Jones et al. [1998] in a noise-free setting. Their framework has latter been extended to optimisation of the conditional expectation of a stochastic black box [see e.g. Frazier et al., 2009, Srinivas et al., 2009, Picheny et al., 2013]. Recently, strategies optimising risk measures have been proposed, In

*Accepted for the 38th Conference on Uncertainty in Artificial Intelligence* (UAI 2022).

particular, Cakmak et al. [2020], Nguyen et al. [2021b,a] proposed new algorithms to optimise for the quantile and CVaR, but for a different use case where the space $\Omega$ is actually controllable and quantitative (e.g., $\omega$ is a random parameter of the black-box with a Gaussian distribution). This use case generally facilitates BO, as modelling can be performed over $\Psi$ over the joint $\mathcal{X} \times \Omega$ space ($g$ being then inferred from $\Psi$), for which observations are directly available.

When $\Omega$ is not controllable, one solution is to assume a parametric distribution (e.g. Gaussian) for $Y_x$, and infer its mean and variance as a function of $x$ ($g$ being then obtained from the inferred distribution of $Y_x$): see e.g. Kersting et al. [2007], Lázaro-Gredilla and Titsias [2011], Saul et al. [2016], Binois et al. [2018]. However, as we show in our experimental section, such an approach will fail when the choice of the distribution of $Y_x$ is inappropriate. An alternative is to replicate observations (i.e. at a fixed $x$ value) to compute local empirical estimates of $g$, then model $g$ directly from this set of observations. Browne et al. [2016] and Makarova et al. [2021] proposed BO algorithms to optimise quantiles and CVaRs following this principle. However, our experiments also show that intensively repeating observations dramatically hinders the efficiency of BO.

Hence, our approach is based on the following principles: a) we model the risk $g$ directly from noisy observations $Y_x$ of the black box; b) our model does not assume any parametric distribution of $Y_x$; c) our algorithm does not require observation replicates. As quantile optimisation takes substantially more data points that standard BO and as BO incurs a significant computational overhead per step, we focus further on the most likely use-case for quantile BO is in the large batch setting, i.e. where large data volumes can be observed in a relatively small number of optimisation steps.

**Contributions** The contributions of this paper are the following: 1) We propose a new model based on two latent Gaussian Processes (GPs) to estimate quantiles or expectiles that is tailored to heteroscedastic noise. 2) We use sparse posterior and variational inference to support potentially large datasets. 3) We propose a new Bayesian algorithm suited to optimise conditional quantiles or expectiles in a data efficient manner. Two batch-sequential acquisition strategies are designed to find a good trade-off between exploration and intensification. 4) The ability of our algorithm to optimise quantiles is illustrated on multiple test problems.

## 2 BAYESIAN METAMODELS OF RISK MEASURES

For a given input point $x$, the quantile of order $\tau \in (0,1)$ of $Y_x$ can be defined as

$$g(x) = q_\tau(x) = \underset{q \in \mathbb{R}}{\arg\min} \, \mathbb{E}\big[l_\tau(Y_x - q)\big], \qquad (1)$$

where $l_\tau$ is the pinball loss [Koenker and Bassett Jr, 1978]:

$$l_\tau(\xi) = (\tau - \mathbb{1}_{(\xi<0)})\xi, \quad \xi \in \mathbb{R}. \qquad (2)$$

Despite its wide popularity (in particular due to its interpretability), quantiles have some important shortcomings, including not being a coherent measure of risk [Artzner et al., 1999]. As an alternative, Newey and Powell [1987] introduced the expectile as the minimiser of an asymmetric quadratic loss:

$$e_\tau(x) = \underset{q \in \mathbb{R}}{\arg\min} \, \mathbb{E}\big[l_\tau^e(Y_x - q)\big], \qquad (3)$$

$$l_\tau^e(\xi) = |\tau - \mathbb{1}_{(\xi<0)}|\xi^2, \quad \xi \in \mathbb{R}. \qquad (4)$$

Contrary to quantiles, expectiles depend on the entire distribution and are a coherent measure of risk. Their main drawback is their lack of interpretability [see Waltrup et al., 2015, for a discussion].

We detail in the next section how these losses can be used to get an estimate of the objective function $g(x)$ using a dataset $\mathcal{D}_n = \big((x_1, y_1) \cdots, (x_n, y_n)\big) = (\mathcal{X}_n, \mathcal{Y}_n)$ that does not necessarily require replicates of observations at the same input location.

### 2.1 QUANTILE AND EXPECTILE METAMODEL

Different metamodels have been proposed to estimate a quantile function, such as artificial neural networks [Cannon, 2011], random forest [Meinshausen, 2006] or nonparametric estimation in reproducing kernel Hilbert spaces [Takeuchi et al., 2006]. While the literature on expectile regression is less extended, neural network [Jiang et al., 2017] or SVM-like approaches [Farooq and Steinwart, 2017] have been developed as well. All the approaches cited above defined an estimator of $g$ as the function that minimises (optionally with a regularisation term)

$$\mathcal{R}_e[g] = \frac{1}{n} \sum_{i=1}^n l\big(y_i - g(x_i)\big), \qquad (5)$$

with $l = l_\tau$ for the quantile estimation and $l = l_\tau^e$ for the expectile. Intuitively, minimising (5) is equivalent (asymptotically) to minimising (1) or (3).

These approaches however share a common drawback: they do not capture the uncertainty associated with each prediction. This is a significant problem in our setting since quantifying this uncertainty is of paramount importance to define the exploration/intensification trade-off. This limitation can be overcome by using a probabilistic model such as

$$y = g(x) + \epsilon(x),$$

where $g$ is either an unknown parametric function [Yu and Moyeed, 2001] or a Gaussian process [Boukouvalas et al.,

2012, Abeywardana and Ramos, 2015], and where the distribution of $\epsilon$ depends on the quantity to be estimated. For modelling a quantile, $\epsilon$ should follow an asymmetric Laplace distribution:

$$p_\epsilon(e) = \frac{\tau(1-\tau)}{\sigma} \exp\left(-\frac{l_\tau(e)}{\sigma}\right).$$

For approximating an expectile, one may use the asymmetric Gaussian distribution:

$$p_\epsilon(e) = C(\tau, \sigma) \exp\left(-\frac{l_\tau^e(e))}{2\sigma^2}\right), \tag{6}$$

with $C(\tau, \sigma) = \dfrac{\sqrt{2\tau(1-\tau)}}{\sigma\sqrt{\pi}(\sqrt{\tau} + \sqrt{1-\tau})}$.

In both cases, the associated likelihood is given by

$$p(\mathcal{Y}_n | g) = \prod_{i=1}^{n} p_\epsilon(y_i - g(x_i)). \tag{7}$$

As this likelihood is a monotonic transformation of the empirical risk associated to the pinball or asymmetric quadratic loss (5), their minimisers coincide.

Although the Bayesian quantile model presented above is well known, the Bayesian expectile model we just introduced is new to the best of our knowledge. It is worth noting that the non-conjugacy between the prior on $g$ and the likelihood functions implies that the posterior distribution of $g$ given the data is not available in closed form. To overcome this, Boukouvalas et al. [2012] use expectation propagation whereas Abeywardana and Ramos [2015] favours variational inference. The latter appears to be one of the most competitive approaches on the benchmark presented in Torossian et al. [2019] so we will embrace the variational inference framework in the remainder of the paper.

One limitation of the aforementioned methods is that they can result in overconfident predictions in heteroscedastic settings, as illustrated in Figure 1. The main reason is that they only use a single parameter $\sigma$ to capture the spread for the likelihood function, which amounts to enforcing that the noise amplitude does not change over the input space. We believe this can be a severe limitation in the context of quantile optimisation since the fluctuation of the quantile value over the input space is likely to be dictated by the noise distribution itself not being stationary.

To overcome this issue, we propose to build quantile and expectile models where the spread of the asymmetric Laplace and Gaussian likelihoods varies across the input space. For both distributions, this additional flexibility can be achieved by redefining $\sigma$ in equations 7 and 6 as a function of the input parameters. Intuitively, a small value of $\sigma(x)$ means that there is a high penalty for having an estimate of $g(x)$ that is far away from the data, whereas a large value of $\sigma(x)$ means that this penalty is limited and thus leads to more

regularity in the model predictions. In practice, we choose a Gaussian prior for $g$ and a log-Gaussian prior for $\sigma$,

$$g(x) \sim \mathcal{GP}\big(\mu_g(x), k_\theta^g(x, x')\big), \tag{8}$$

$$\log \sigma(x) \sim \mathcal{GP}\big(\mu_\sigma(x), k_\theta^\sigma(x, x')\big). \tag{9}$$

GPs are very popular surrogate models in BO due to their flexibility at modelling smooth functions. Smooth objective functions also have smooth quantiles and so GPs are also a natural choice when modelling quantiles. The choice of a log GP prior for $\sigma$ is simply to ensure that the Laplace variance takes only positive values. Our quantile model can be compared to the Heteroscedastic GP model introduced by Saul et al. [2016], but with a different likelihood function so that the posterior mode corresponds to a quantile or an expectile.

## 2.2 INFERENCE PROCEDURE

Although in most situations one can obtain a reasonable estimate of a mean value using only a handful of samples, inferring low or high order quantiles or expectiles tends to require a much larger number of observations, since they require information associated to the tails of the distribution. The inference procedure for the proposed probabilistic model must thus be able to cope with relatively large datasets, with the number of observations likely in the order of a few hundreds to a million data points.

A well established method that supports both large datasets and non-conjugate likelihoods is the Sparse Variational GP framework [SVGP, Titsias, 2009, Hensman et al., 2013]. We briefly expose here the basic principles behind SVGP, and defer the interested reader to the reference above or the recent tutorial of Leibfried et al. [2020] for more details.

The SVGP framework consists in approximating the intractable or computationally expensive posterior distribution $p(g, \sigma | \mathcal{Y}_n)$ by a distribution $p(g, \sigma | g(\boldsymbol{Z}) = \boldsymbol{u}_g, \sigma(\boldsymbol{Z}) = \boldsymbol{u}_\sigma)$, where $\boldsymbol{Z} \in \mathcal{X}^N$ and $\boldsymbol{u}_g$, $\boldsymbol{u}_\sigma$ are $N$-dimensional random variables:

$$\boldsymbol{u}_g \sim \mathcal{N}(\boldsymbol{u}_g | \mu_g, S_g) \text{ and } \boldsymbol{u}_\sigma \sim \mathcal{N}(\boldsymbol{u}_\sigma | \mu_\sigma, S_\sigma).$$

The parameters $\boldsymbol{Z}$, $\mu_g$, $S_g$, $\mu_\sigma$, $S_\sigma$, are referred to as the variational parameters. The $\boldsymbol{Z}$'s are often called *inducing points*. Intuitively, $\boldsymbol{u}_g$ are random variables that act as pseudo-observations at the locations $\boldsymbol{Z}$. Those locations are either pre-determined (taken e.g. as a random subset of the data or as the centroids returned by a k-means algorithm) or optimised.

The variational parameters can be optimised jointly with the model parameters (e.g. mean function coefficients or kernel hyperparameters) such that Kullback-Leibler divergence between the approximate and the true posterior is as small as possible. In practice, this is achieved by maximising the

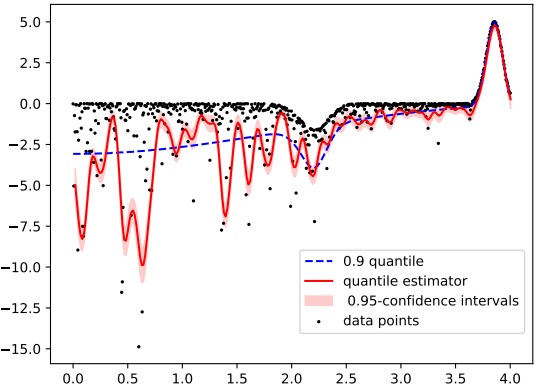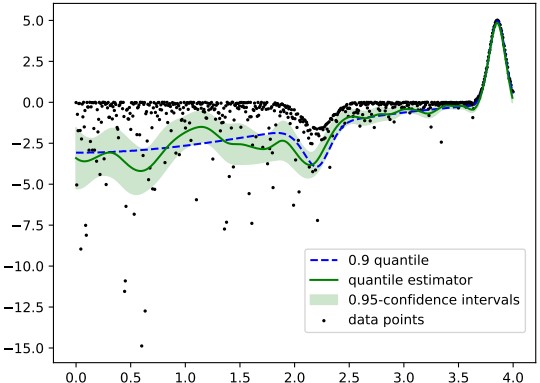

Figure 1: GP quantile model from Abeywardana and Ramos [2015] (left) and ours (right) on data with high heteroscedasticity. The left model cannot compromise between very small observation variances around $x = 4$ and very large variances ($x \leq 2$), largely overfits on half of the domain and returns overconfident confidence intervals. In contrast, our model captures both the low and high variance regions, while returning well-calibrated confidence intervals.

Evidence Lower Bound (ELBO):

$$\sum_{i=1}^{n} \int \log p(y_i|g_i, \sigma_i) \tilde{p}(g_i)\tilde{p}(\sigma_i)dg_i d\sigma_i$$
$$- \text{kl}\left(\tilde{p}(u_q)||p(u_q)\right) - \text{kl}\left(\tilde{p}(u_\sigma)||p(u_\sigma)\right),$$

where $\tilde{p}(g_i)$ and $\tilde{p}(\sigma_i)$ are shorthands for the variational posterior distributions at $x_i$:

$$\tilde{p}(g_i) = \int p(g(x_i)|g(\boldsymbol{Z}) = \boldsymbol{u}_g)p(u_g)du_g$$
$$= \mathcal{N}(g_i|K_{x_i,u_g}K_{u_g,u_g}^{-1}\mu_g, K_{x_i,x_i} + Q_g),$$

where $Q_g = K_{x_i,u_g}K_{u_g,u_g}^{-1}(S_g - K_{u_g,u_g})K_{u_g,u_g}^{-1}K_{u_g,x_i}$.

To optimise the ELBO in practice, Hensman et al. [2013] proposed a numerical solution allowing for mini-batching and the use of stochastic gradient descent algorithms such as Adam [Kingma and Ba, 2014][1].

## 3 BAYESIAN OPTIMISATION

Classical BO algorithms work as follow. Firstly, a posterior distribution on $g$ is inferred from an initial set of experiments $\mathcal{D}_n$ (typically obtained using a space-filling design). Then the next input point to evaluate is chosen as the maximiser of an *acquisition function* $\alpha_n : \mathcal{X} \rightarrow \mathbb{R}$, computed using the posterior of our surrogate model(s). The objective function is sampled at the chosen input and the posterior on $g$ is updated. These steps are repeated until the budget is exhausted. The efficiency of such strategies depends on how informative the posterior distribution of $g$ is but

also on the exploration/exploitation trade-off provided by the acquisition function. Many acquisition functions have been designed to control this trade off, among them the *Expected improvement* [EI, Jones et al., 1998], *upper confidence bound* [UCB, Srinivas et al., 2009], *knowledge gradient* [KG, Frazier et al., 2009] and *Entropy search* [ES, Hennig and Schuler, 2012].

In the case of quantiles and expectiles, adding points one at a time is impractical since many points are typically necessary to induce a significant change in the posterior for $g$. Hence, we focus here on *batch-BO* strategies, for which the acquisition recommends a batch of $B > 1$ points instead of a single one. The above-mentioned acquisition functions have been extended to handle batches: see for instance Marmin et al. [2015] for EI, [Wu and Frazier, 2016] for KG, or Desautels et al. [2014] for UCB. However, none of these approaches actually fit our settings for two main reasons. Firstly, most parallel acquisitions ( like those based on EI or KG) make use of explicit update equations for the GP moments and assume access to a Gaussian posterior for observations, neither of which are available for our quantile (or expectile) model. For example, expected improvement assumes that we see noisy observations of the object that we wish to predict the improvement of. This is not the case in the quantile optimisation setting, as our observations are not noisy realisations of the quantile. Secondly, most existing batch acquisitions are designed for small batches (say, $B \leq 5$) and become numerically intractable for the larger batches (say, $B > 50$) that provide the data volumes necessary for optimising quantiles and expectiles.

We now propose the first acquisition functions that can be applied to our quantile GP surrogate model, one based on Thompson sampling and one on max-value entropy search.

---

[1]first order optimizers such as Adam are particularly relevant for the quantile as that can handle the non-differentiability of the objective function due to the non-differentiability of the pinball loss at the origin.

## 3.1 THOMPSON SAMPLING

Thompson sampling (TS) is becoming increasingly popular in BO, in particular because of its embarrassingly parallel nature allowing full scalability with the batch size [Hernández-Lobato et al., 2017, Kandasamy et al., 2018, Vakili et al., 2021].

Given the posterior on $g$, an intuitive approach is to sample $\Psi(x, .)$ according to the probability that $x$ is the location of the maximum of $g$. Despite this distribution usually being intractable, one may achieve the same result by sampling from the posterior of $g$ and then selecting the input that corresponds to the maximiser of the sample. Such approach directly extends to batches of inputs, by drawing several samples and selecting all the maximisers.

The main drawback of GP-based TS is the cost of sampling, which can only be done exactly at a finite number of input locations and with cubic cost in the number of locations. An alternative is to rely on a finite rank approximation of the kernel, but this has been found to have an undesirable effect known as *variance starvation* [Wang et al., 2018].

Wilson et al. [2020] showed that pairing sparse GP models with the so-called *decoupled sampling* formulation avoids the variance starvation issue. Vakili et al. [2021] then demonstrated that such an approach delivered excellent empirical performance on high noise, large budget, large batch scenarios, while enjoying the same theoretical guarantees as the vanilla TS approach. Here, we build upon Vakili et al. [2021], and apply their algorithm to the variational posterior of $g$ to obtain draws directly from the quantile or expectile model. The posterior over $\sigma$, which controls the observation noise, is not used during the TS algorithm.

The procedure for generating quantile samples from the variational posterior of $g$ can be summarised as follows: First, a continuous sample from the prior of $g$ is generated using Random Fourier Features (see supplementary material). Second we sample from the inducing variables $u_g$. Third, we compute the mean function $m(x)$ of a GPR model that interpolates the dataset $\{Z, u_g - s(Z)\}$. Finally, the posterior sample is obtained by correcting the prior samples with the mean function $v(x) = s(x) + m(x)$.

## 3.2 INFORMATION-THEORETIC QUANTILE OPTIMISATION WITH GIBBON

Another particularly intuitive search strategy for BO is to choose the evaluations that will maximally reduce the uncertainty in the minimiser of the objective, an approach known as max-value entropy search [MES, Wang and Jegelka, 2017]. For quantile optimisation, MES corresponds to reducing uncertainty in the maximal quantile value $g^* = \max_{x \in \mathcal{X}} g(x)$. Following the arguments of Wang and Jegelka [2017], a meaningful measure of uncertainty

reduction in this context is taken as the gain in mutual information between a set of candidate evaluations and $g^*$ [see Cover and Thomas, 2012, for an introduction to information theory]. Principled information-theoretic optimisation then corresponds to finding batches of $B$ input points $\{x_i\}_{i=1}^B$ that maximise

$$\alpha_n(\{x_i\}_{i=1}^B) = \text{MI}(g^*; \{y_{x_i}\}_{i=1}^B | \mathcal{D}_n), \qquad (10)$$

where $y_{x_i}$ are not-yet-observed evaluations of the batch that are estimated with the GP surrogate model.

Although calculating the acquisition function (10) is challenging, there exist effective approximation strategies for GP models with conjugate likelihoods [Moss et al., 2020b, Takeno et al., 2020]. In the remaining of this section we show that the approach used in General-purpose Information Based Bayesian-OptimisatioN [GIBBON, Moss et al., 2021] can be adapted to support asymmetric Laplace or Gaussian likelihood so that information-theoretic acquisition functions can be used for our quantile and expectile models.

Following the derivations of Moss et al. [2021], the application of three well-known information-theoretic inequalities provides the following lower-bound for the mutual information (10):

$$\text{MI}(g^*; \{y_{x_i}\}_{i=1}^B | \mathcal{D}_n) \geq \text{H}(\{y_{x_i}\}_{i=1}^B | \mathcal{D}_n)$$
$$- \frac{1}{2} \sum_{i=1}^B \mathbb{E}_{g^*|\mathcal{D}_n} \left[ \log(2\pi e \text{Var}(y_{x_i} | g^*, \mathcal{D}_n)) \right], \quad (11)$$

where $\text{H}(A) = -\mathbb{E}_A [\log p(A)]$ denotes differential entropy. Although calculating the expectation in the second term of (11) is intractable (i.e. no closed-form expression exists for $p(g^* | \mathcal{D}_n)$), we follow another approximation common among information-theoretic acquisition functions and approximate the integral using Monte-Carlo over a set of $M$ sampled maximum values. In particular, we use the Gumbel sampler proposed by Wang and Jegelka [2017], which provides a cheap set of samples $\mathcal{M}_n = \{g_1^*, .., g_M^*\}$ from $p(g^* | \mathcal{D}_n)$.

When calculating the original GIBBON acquisition function, all the terms in the lower bound (11) are tractable, i.e. the conjugacy of their Gaussian likelihood means that $\text{H}(\{y_{x_i}\}_{i=1}^B | \mathcal{D}_n)$ is just the differential entropy of a multivariate Gaussian which, alongside each $\text{Var}(y_{x_i} | g^*, \mathcal{D}_n)$, has a closed-form expression (See Moss et al. [2021] for details). Consequently, this lower bound itself is used as a closed-form approximation to the mutual information. However, in our quantile setting, we no longer have expressions for the first term of (11) — the joint differential entropy of $B$-dimensional variable with a complex correlation structure given by our two latent GPs.

To build an information-theoretic acquisition function suitable for our quantile model, we must apply an additional

approximation. In particular, by using a moment-matching approximation, we can replace the intractable joint differential entropy with the differential entropy of a multivariate Gaussian of the same covariance, leading to our proposed Quantile GIBBON (Q-GIBBON) acquisition function;

$$\alpha_n^{\text{Q-GIBBON}} = \frac{1}{2} \log |C| - \frac{1}{2M} \sum_{g^* \in \mathcal{M}_n} \sum_{i=1}^{B} \log V_i(g^*),$$

(12)

where $|C|$ is the determinant of the $B \times B$ predictive covariance matrix with elements $C_{i,j} = \text{Cov}(y_{x_i}, y_{x_j})$ and $V(g^*)$ denotes the conditional variances, $V_i(g^*) = \text{Var}(y_{x_i}|g^*, \mathcal{D}_n)$. Crucially, all the terms of Q-GIBBON have closed-form expressions (see supplementary material for a derivation of $C$ and $V$ from our quantile GP).

Although applying an additional moment-matching approximation means that Q-GIBBON is no longer a lower bound on the true mutual information, we found that it provides very efficient optimisation (see Section 4). In fact, we tried much more expensive but unbiased Monte-Carlo approximations which did not result in noticeable difference in performance.

In practice, directly searching for the set of $B$ points that maximise $\alpha_n^{\text{Q-GIBBON}}$ is a very challenging task, due to the dimensionality ($B \times D$) and multimodality of the acquisition function. However, the Q-GIBBON formulation makes it particularly well-suited for a *greedy* approach, where we first optimise Q-GIBBON for $B = 1$, then optimise for $B = 2$ while fixing the first point to the previously found value, etc. until $B$ points are found. Note that, just like with the standard GIBBON acquisition function, the diversity between the elements in batches is provided by a repulsion term (i.e. the first term of (12)) that depends only on the correlation of the points in the batch. This is in contrast to other greedy batch methods like the Kriging Believer of Ginsbourger et al. [2010] or the one-shot knowledge gradient Balandat et al. [2019] which are unsuitable for the large batch setting as they require updates to their surrogate model's posterior as we add each new batch element.

# 4 EXPERIMENTS

We now evaluate our proposed model and acquisition functions on a set of synthetic tasks and two real-world optimisation problems. All that follows could equivalently be applied to expectiles, experiments are focused on quantile optimisation to streamline the exposition.

## 4.1 ALGORITHM BASELINES

To our knowledge, there is no other existing BO algorithm dedicated to optimising quantiles in our considered setting.

The most similar algorithms are those of Cakmak et al. [2020] and Makarova et al. [2021]. However, Cakmak et al. [2020] requires precise control over the noise generation process, while Makarova et al. [2021] seek to find solutions with low levels of observation noise but do not provide a method for optimising a specific quantile level.

We can, however, apply standard BO methods to perform quantile optimisation if direct observations of the quantiles are available. This is achievable by using repeated observations, which allows computing a (pointwise) empirical quantile. As direct observations are available, a standard GP Regression model (GPR) can be used to provide a posterior on $g$ [Plumlee and Tuo, 2014]. One can also bootstrap the repeated observations to obtain variance estimates of the empirical quantiles, to improve further the model by accounting for varying observation noise. Next, a BO procedure can be defined based on any classical acquisition function. Here we choose the vanilla EI. With this strategy, each batch consists of a single point in the input space, repeated a number of times. In the following we denote this baseline GPR-EI.

Our second baseline is based on the model of Saul et al. [2016]. This model has a classic approach to heteroscedastic noise [see similar approaches e.g. in Kersting et al., 2007, Lázaro-Gredilla and Titsias, 2011, Binois et al., 2018], where one (latent) GP is used to represent the mean of the observations, and another GP to represent the log of the noise variance, assuming that the noise is Gaussian. This model uses the same variational framework as our quantile model. With this model, estimation is focused on the mean and variance of the observations, and the quantile prediction is obtained using the Gaussian quantiles. We used this model with Thompson sampling, allowing batch acquisitions without repetitions. In the following we denote this baseline HetGP.

## 4.2 IMPLEMENTATION

All models are built using the `gpflux` library [Dutordoir et al., 2021], and our BO procedure uses `trieste` [Berkeley et al., 2022]. All models use a Matern 5/2 kernel, and all acquisition functions (or GP samples in the case of TS) are optimised using a multi-start BFGS scheme.

Our quantile model requires a design choice for the inducing points placement. We follow the findings of Vakili et al. [2021] and reinitialise their placement for each model fit using the centroids of a k-means procedure on the data points. This tends to concentrate the inducing points near the optimal areas as more data is collected by BO and offer a better local expressivity in those areas. Our implementation of decoupled Thompson sampling uses $1,000$ random Fourier features (see supplementary material for detailed expressions). To sample minimum values for Q-GIBBON we use the Gumbel sampler of Wang and Jegelka [2017]

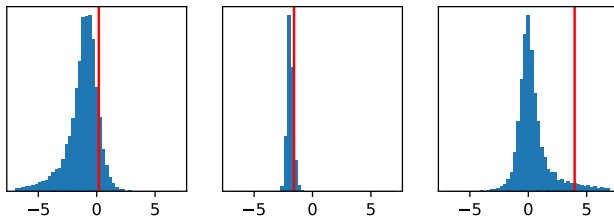

Figure 2: Examples of marginal distributions for one GLD-based problem at three different locations of the input space. The vertical lines show the 95% quantiles.

with $10,000 \times D$ random initial points.

## 4.3 SYNTHETIC PROBLEMS

**Problem description**  We generated a set of synthetic problems based on the Generalised Lambda Distribution [GLD, Freimer et al., 1988], a highly flexible four-parameter probability distribution function designed to approximate several well-known parametric distributions. The four parameters define the location, scale, left and right shape of the distribution, respectively. By varying the value of each parameter as a function of $x$, one can create a black-box with high noise, heteroscedasticity and non-Gaussianity:

$$Y_x \sim GLD(\lambda_1(x), \dots, \lambda_4(x))). \tag{13}$$

To generate a large set of problems with varying dimensionality while controlling the multimodality of the problem at hand, we used GP random draws for the $\lambda_i$'s. See supplementary material for a full description. Figure 2 shows examples of marginal distributions (for different $x$ values) for one such problem.

We consider two input space dimensions: $D = 3$ and 6 and two quantile levels, $\tau = 0.75$ and $0.95$. We use as an initial budget $50D$ observations[2], uniformly distributed across the input space and a total budget of $250D$ observations, acquired in batches of either $B = 10$ or 50 points. Each strategy is run on 50 different problems. We report here the simple regret in Figure 3, averaged over the 50 problems, with confidence intervals.

**Results**  In almost all cases, our approaches largely outperform the two baselines, the exception being on the simpler problem (small dimension and batch size) for which the GPR baseline is comparable to TS (GIBBON being substantially better for the 0.75 quantile). The HetGP approach sometimes deliver good early performance (e.g. dimension 6, batch 10, 75% quantile), beating the GPR baseline but not our quantile approaches. However, its performance is considerably hindered by the normality assumption of the

[2]this number is relatively large to allow initialising GPR-EI with several distinct input points, each with $B$ repetitions.

noise (as would any approach assuming a parametric distribution). This is particularly visible during later iterations when the optimum is more precisely identified, the model wrongly assumes that the noise is Gaussian and we often see the regret increasing. Comparing acquisition strategies, GIBBON clearly outperforms TS for $D = 3$. In dimension 6, both approaches are roughly comparable.

## 4.4 LUNAR LANDER

**Problem description**  The Lunar Lander problem is a popular benchmark for noisy BO [Moss et al., 2020a, Eriksson et al., 2019]. In this well-known reinforcement learning task, we must control three engines (left, main and right) to successfully land a rocket. The learning environment and a hard-coded PID controller is provided in the OpenAI gym.[3] We seek to optimise 6 thresholds present in the description of the controller to provide the largest expected reward: finding those thresholds defines the BO task. Our RL environment is exactly as provided by OpenAI. We lose 0.3 points per second of fuel use and 100 if we crash. We gain 10 points each time a leg makes contact with the ground, 100 points for any successful landing, and 200 points for a successful landing in the specified landing zone. Each individual run of the environment allows the testing of a controller on a specific random seed.

This problem is particularly well-suited for a quantile approach, since reward is stochastic, highly non-Gaussian, and the landing problem is a clear case for which one would want guarantees against risk. Moreover, as certain configurations of the lunar lander lead to much less stable behaviour and a greater range of outcomes, this problem requires heteroscedastic models.

**Results**  For this problem, we ran each algorithm 10 times (starting from different initial conditions), with batches of $B = 25$ points, 300 initial observations and $1,500$ in total. We aim to maximise respectively the 2% and 10% quantile of the reward. Due to the high cost of calculating the true quantiles of the lunar lander experiment (i.e. they must be calculated empirically across a large collection of runs), we only report the reward quantile obtained after half and all the iterations (see Table 1) and only run one of our two proposed acquisition functions. We choose TS over GIBBON as our synthetic GLD experiments suggest that TS outperforms Q-GIBBON on problems with larger (i.e. 6) dimensions.

Here, the HetGP approach completely fails at recommending good solutions, which can be explained by the strong violation of Gaussianity of the noise. TS from our Quantile GP substantially outperforms GPR-EI, achieving higher performance with much lower variability, both at intermediate and final steps.

[3]*https://gym.openai.com/*

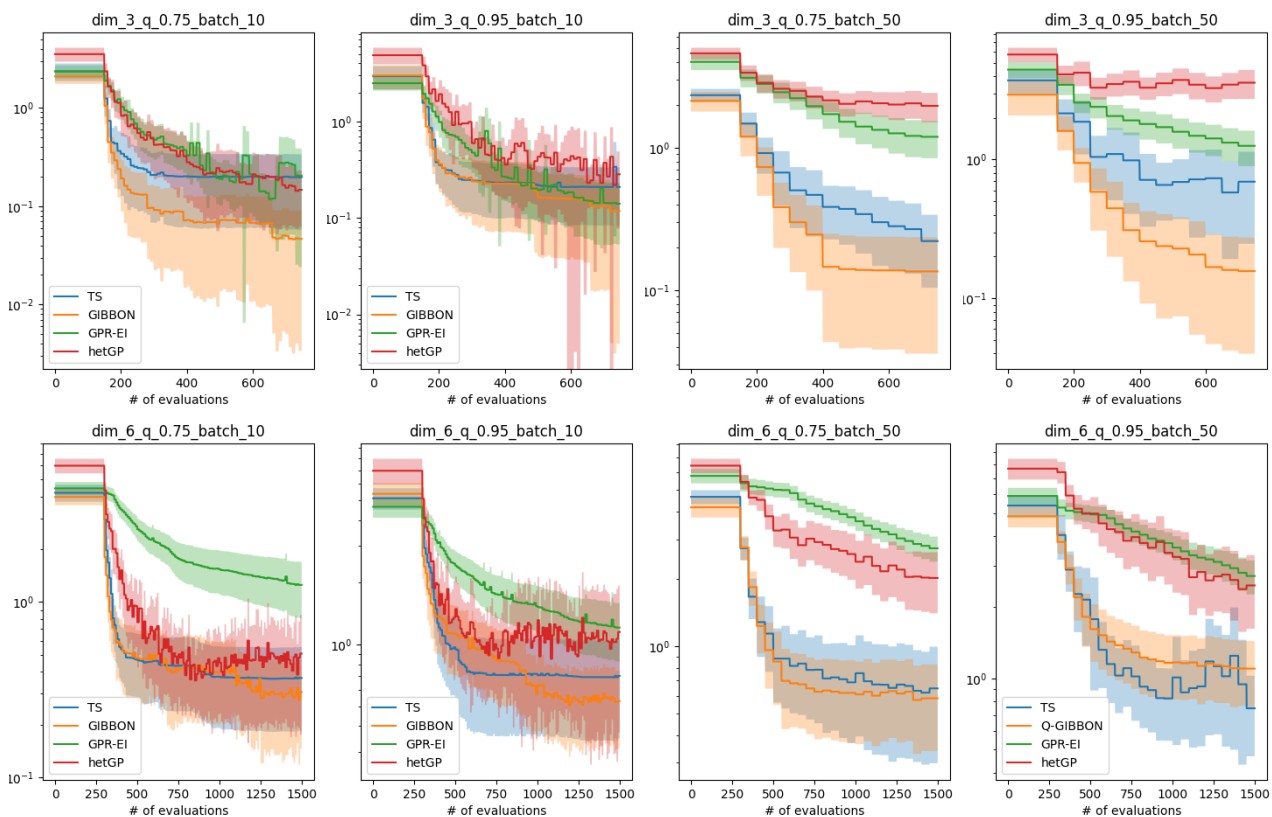

Figure 3: The mean and 95% confidence intervals of regret on synthetic problems in dimension 3 (top) and 6 (bottom), for two quantile levels ($\tau = 0.75, 0.95$) and medium ($B = 10$, left) and large ($B = 50$, right) batch sizes.

|       |         | 750 obs        | 1500 obs       |
|-------|---------|----------------|----------------|
| 10%   | GPR-EI  | 94.6 (106.1)   | 159.5 (110.9)  |
|       | HetGP   | -38.7 (17.5)   | -18.7 (4.2)    |
|       | TS      | 204.3 (53.8)   | 255.2 (8.0)    |
| 2%    | GPR-EI: | 141.8 (82.9)   | 187.4 (80.6)   |
|       | HetGP   | -35.1 (31.9)   | -19.2 (25.2)   |
|       | TS      | 193.8 (56.4)   | 238.9 (14.6)   |

Table 1: Mean and standard deviation over 10 runs for the 10% and 2% quantiles of the reward on the lunar lander problem.

## 4.5   LASER TUNING

**Problem Description**   For our final experiment, we test our quantile optimisation in a real-world setting inspired by the Free-Electron Laser (FEL) tuning example of McIntire et al. [2016]. This is a challenging 16-dimensional optimisation task where we must configure the strengths of magnets manipulating the shape of the FEL's electron beam, seeking to build a powerful and stable beam suitable for use in scientific experiments. Due to the high levels of observation noise in this problem and as stability of the resulting beam is of critical importance for conducting reliable experiments, it is clearly beneficial to encode a level of risk-adversity into

the optimisation. Therefore, there are clear advantages for using quantile optimisation for FEL calibration.

As we do not have access to the FEL directly, we follow McIntire et al. [2016] and use their $4,074$ observed X-ray pulse energy measurements to build GP surrogates from which we can simulate pulse energy at any new magnet configuration. To simulate the effect of observation noise, McIntire et al. [2016] add additional Gaussian perturbations to the simulated values. However, we found that the noise in this system was actually skew Gaussian and varied in scale and skew across the search space. Consequently, we simulate observation noise from a skew Gaussian distribution with location, scale and shape parameters also modelled with additional GPs (i.e. a setup similar to our GLD examples). As many of the $4,074$ energy measurements are evaluated at very similar input locations, rounding these inputs to four decimal places provides us with many repeated evaluations, allowing the empirical estimation of each parameter of the skew Gaussian distribution at each of these inputs. The location, scale and shape GPs are then determined to predict the parameters of the skew Gaussian noise distributions for any candidate magnet configuration.

**Results**   Figure 4 shows the performance of each algorithm over 10 repetitions, seeking to maximise the 30% quantile

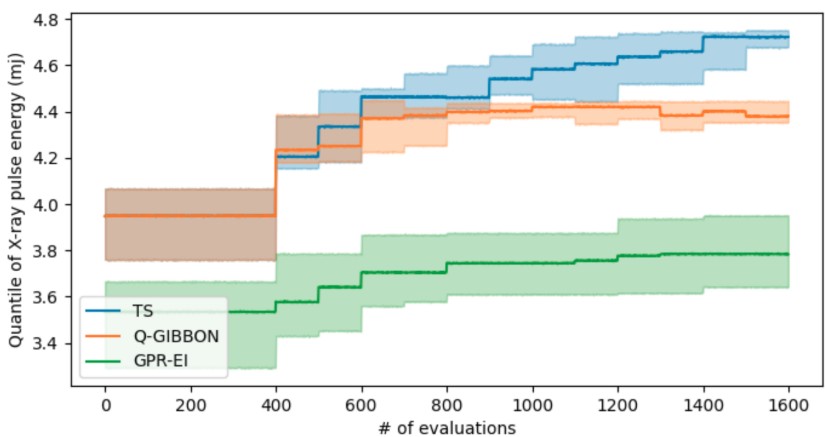

Figure 4: The mean and $95\%$ confidence intervals of best 0.3 quantile found across 10 repetitions of the FEL tuning task.

of pulse energy. The models are initialised with 400 data points randomly chosen from the full dataset, and a further 1,200 points are collected with BO in batches of 100 points. Our algorithms based on quantile GP models substantially outperform the replicate-based GPR baseline. In fact, by using TS with a quantile GP, we are able to find solutions very close to the optimal value (4.8). We hypothesise that the relatively poor performance of our Q-GIBBON acquisition function is due to the high dimension of this problem. The Gumbel sampler used by Q-GIBBON for sampling minimal-values is based on random sampling and so its performance likely degrades as the input dimension increases. Since the performance of information-theoretic BO is sensitive to the quality of these samples [Moss et al., 2021], extending information-theoretic BO to high dimensional problems like FEL tuning remains an open question.

## 4.6 CONCLUDING COMMENTS

We have presented a new setting to estimate quantiles and expectiles of stochastic black box functions that is well suited to heteroscedastic cases. We then used the proposed model to create two BO algorithms designed for the optimisation of conditional quantiles and expectiles without repetitions in the experimental design. These algorithms outperform the state of the art on several test problems with different dimensions, quantile orders, budgets and batch sizes.

Overall, our experiments clearly show that the performance gap between our approaches and the GPR-EI baseline increases with the batch size and problem dimension. Since GPR-EI relies on repetitions, it is much more limited in terms of exploration, while our approaches can evaluate $B$ unique points at each BO iteration. Hence, our approach is much less sensitive to the curse of dimensionality.

Experiments also show that for low-dimensional, smaller batches, Q-GIBBON is the best alternative, while with increasing dimension and batch size, the simpler Thompson

sampling seems to perform best. Depending on the available hardware, the parallel nature of TS might also provide substantial advantages in terms of wall-clock time.

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
