# OpenReview forum: "Bayesian Quantile and Expectile Optimisation"
_auai.org/UAI/2022/Conference — UAI 2022 Poster_

### Official Review · Reviewer_FPqD · 2022-04-06

**Q2(1) Originality/Novelty:** 3
**Q2(2) Significance/Impact:** 2
**Q2(3) Correctness/Technical Quality:** 3
**Q2(6) Clarity Of Writing:** 3
**Q6 Overall Score:** 5
**Q8 Confidence In Your Score:** 3

**Q1 Summary And Contributions:**

The paper proposes a method for Bayesian quantile and expectile optimization which shifts the focus of BO from a mean estimate to a quantile (or expectile) of the estimated objective function. This provides a more robust method with respect to extreme events. The proposed method handles heteroskedasticity by using two latent GPs and large datasets with sparse inducing points approximations. Finally the authors propose two adaptations of classical acquisition functions to the quantile case.

**Q2 Assessment Of The Paper:**

More detailed information regarding each of these aspects is given below:

**Q2(4) Quality Of Experiments (Optional):**

2: Fair: The experimental evaluation is weak: important baselines are missing, or the results do not adequately support the main claims.

**Q2(5) Reproducibility:**

2: Fair: Key resources (e.g., proofs, code, data) are unavailable but key details (e.g., proof sketches, experimental setup) are sufficiently well-described for an expert to confidently reproduce the main results.

**Q3 Main Strengths:**

- The idea of using directly on quantile/expectiles is very important in several applications and it is good to see work in this direction
- The paper is well written and easy to follow
- the experimental section is convincing

**Q4 Main Weakness:**

- in the experiments the authors only compare against a standard EI, it would be interesting to compare a GP model with a quantile based acquisition function such as expected quantile improvement
- The greedy implementation of the Q-GIBBON acquisition function seem to cause some issues when batch-size B is quite large (see bottom right plot of fig.3).

**Q5 Detailed Comments To The Authors:**

- How would a strategy such as expected quantile improvement (with a standard GP), see Picheny et al. 2013, work here? Why haven’t the authors considered it?

- I am wondering if a more traditional modelling of heteroskedastic noise would also work here. For example would a GP model as in Binois et al. 2018 work here? Could the authors comment on extending the proposed acquisition functions to such method?

- For the greedy nature of the Q-GIBBON optimization, would it be possible to choose greedily 2 points at once (by using a 2-dimensional Q-GIBBON)? Would this reduce the issues due to the greedy optimization?

Picheny, V., Ginsbourger, D., Richet, Y., & Caplin, G. (2013). Quantile-based optimization of noisy computer experiments with tunable precision. *Technometrics*, *55*(1), 2–13.

Binois, M., Gramacy, R. B., & Ludkovski, M. (2018). Practical Heteroscedastic Gaussian Process Modeling for Large Simulation Experiments. *Journal of Computational and Graphical Statistics*, *27*(4), 808–821.

**Q7 Justification For Your Score:**

The paper  presents an interesting method in a clear way. I think the paper has value, however the experimental section only compares against the classic EI. Moreover I think some ablation studies on the choices made in the paper could be useful to understand the behavior of the method.

**Q9 Complying With Reviewing Instructions:**

1: Yes.

---

### Official Review · Reviewer_Yd6g · 2022-04-07

**Q2(1) Originality/Novelty:** 3
**Q2(2) Significance/Impact:** 2
**Q2(3) Correctness/Technical Quality:** 3
**Q2(6) Clarity Of Writing:** 4
**Q6 Overall Score:** 6
**Q8 Confidence In Your Score:** 3

**Q1 Summary And Contributions:**

The paper proposes novel variational models for Bayesian quantile and expectile regression (based on two latent Gaussian processes) which can deal with heteroscedastic noise. An inference algorithm based on existing methods (space VI) is designed to deal with high-dimensional data. A novel algorithm is proposed to optimize conditional quantiles and expectiles. The proposed method is evaluated on multiple interesting data sets.

**Q10 Ethical Concerns (Optional):**

None.

**Q2 Assessment Of The Paper:**

More detailed information regarding each of these aspects is given below:

**Q2(4) Quality Of Experiments (Optional):**

3: Good: The experimental evaluation is adequate, and the results convincingly support the main claims.

**Q2(5) Reproducibility:**

2: Fair: Key resources (e.g., proofs, code, data) are unavailable but key details (e.g., proof sketches, experimental setup) are sufficiently well-described for an expert to confidently reproduce the main results.

**Q3 Main Strengths:**

The paper is well-structured and clearly written. The performance of the proposed method is evaluated on interesting data sets and the results have insightful conclusions and demonstrate the solid performance of the proposed method.

**Q4 Main Weakness:**

I do not see any obvious weaknesses.

**Q5 Detailed Comments To The Authors:**

No code was submitted for reproducibility, but as far as I understand from the paper it will be shared on GitHub later?

**Q7 Justification For Your Score:**

The paper provides a novel model for Bayesian quantile and expectiles regression (with the clear motivation of its relevance), incorporates existing approaches into the general framework (sparse VI), and develops a new Bayesian algorithm to optimize conditional quantiles and expectiles. The experiments are interesting and demonstrate good performance in multiple problems. All in all, it is a solid paper with clear motivation.

**Q9 Complying With Reviewing Instructions:**

1: Yes.

---

### Official Review · Reviewer_eNVT · 2022-04-12

**Q2(1) Originality/Novelty:** 3
**Q2(2) Significance/Impact:** 3
**Q2(3) Correctness/Technical Quality:** 3
**Q2(6) Clarity Of Writing:** 3
**Q6 Overall Score:** 6
**Q8 Confidence In Your Score:** 2

**Q1 Summary And Contributions:**

Most Bayesian Optimization algorithms optimize the expected value of an unknown function. In risk-averse settings, it is advantageous to instead optimize for the quantile. The paper describes a novel algorithm for achieving this goal. Unlike current algorithms, the proposed algorithm does not require repeated sampling for the same input points to estimate the quantile. Instead, the new GP-based model estimates quantiles and expectiles directly, which is supported by experiments.

**Q2 Assessment Of The Paper:**

More detailed information regarding each of these aspects is given below:

**Q2(4) Quality Of Experiments (Optional):**

3: Good: The experimental evaluation is adequate, and the results convincingly support the main claims.

**Q2(5) Reproducibility:**

3: Good: Key resources (e.g., proofs, code, data) are available and key details (e.g., proofs, experimental setup) are sufficiently well-described for competent researchers to confidently reproduce the main results.

**Q3 Main Strengths:**

The problem posed by the paper appears novel and natural. The paper is clearly written, which makes the ideas easy to follow. The results show clear improvement for the benchmarks studied.

**Q4 Main Weakness:**

The authors focus on 30% and 10% quantiles in their experiments, which are probably considerably easier to estimate than, say, 5% or 1% quantiles. In many real-world problems, people are interested in rare events, so they would want to estimate more "extreme" quantiles. The authors do not give any intuition about how hard that would be - I wonder if this is much harder. The variational inference procedure could have been stated more explicitly.

**Q5 Detailed Comments To The Authors:**

Section 2.2, first paragraph - it is not generally true that obtaining a mean estimate is easier than estimating quantiles - e.g. for "balanced" quantiles of heavy-tailed distributions the median will be considerably more stable than the mean - I think the opening sentence is somewhat misleading.

**Q7 Justification For Your Score:**

The paper presents a new technique on an intuitively important problem, the results look strong though it might be that the experimental setting is "too easy". I do not have a lot of confidence in my assessment as this is far from my area of expertise.

**Q9 Complying With Reviewing Instructions:**

1: Yes.

---

### Official Review · Reviewer_pRZb · 2022-04-13

**Q2(1) Originality/Novelty:** 2
**Q2(2) Significance/Impact:** 2
**Q2(3) Correctness/Technical Quality:** 2
**Q2(6) Clarity Of Writing:** 1
**Q6 Overall Score:** 5
**Q8 Confidence In Your Score:** 4

**Q1 Summary And Contributions:**

This paper firstly proposes variational inference for quantile and expectile regression models and then utilizes Thompson sampling and max-value Thompson sampling for quantile and expectile optimization.

**Q2 Assessment Of The Paper:**

More detailed information regarding each of these aspects is given below:

**Q2(4) Quality Of Experiments (Optional):**

2: Fair: The experimental evaluation is weak: important baselines are missing, or the results do not adequately support the main claims.

**Q2(5) Reproducibility:**

1: Poor: Key details (e.g., proof sketches, experimental setup) are incomplete/unclear, or key resources (e.g., proofs, code, data) are unavailable.

**Q3 Main Strengths:**

This work shows how an existing acquisition function like max-value entropy search and Thompson sampling can be slightly modified for quantile optimization.

This work additionally accounts for heteroscedastic noise.

Different from existing works on quantile optimization, this work does not assume the observation of uncontrolled variables.

**Q4 Main Weakness:**

The main issue with this work is that of clarity of presentation. On the surface, it is easy to read if we are not particular about knowing and understanding the technical details. However, if we want to, the paper lacks considerable technical details for us to verify the correctness and novelty; the authors have chosen to instead replace them with abstract text descriptions. In this case, a reader like myself can only roughly infer what the authors are trying to do instead of knowing for certain. Consequently, the results of this work are not reproducible. See detailed comments for more information.

There is no theoretical performance guarantee for the proposed algorithms.

The experimental results rely on a relatively large number of initial observations. Can the authors show results with fewer initial observations?

The only method of comparison is naive and not competitive empirically. See detailed comments for more information.

**Q5 Detailed Comments To The Authors:**

POST REBUTTAL FEEDBACK

I like to thank the authors for their clarifications. I have a further comment:

7. Since sparse GP is used, isn't it possible to use the matrix inversion lemma to rewrite the NxN predictive covariance matrix into an expression that can be computed in linear time?

I have increased my score by 1 as I think there is some merit to this work. However, I did not increase it further due to significant changes needed to make the details of the technical approach clear for verifying the correctness. After the authors' clarifications, this concern still remains since we can only do so by reviewing the entire final version once again.



PRIOR FEEDBACK

I would strongly recommend that since the experiments do not cover expectile optimization (there is also a lack of discussion on its motivation), the authors can exclude expectile optimization (or defer it to the appendix) and replace with the missing technical details necessary for understanding.

It is not clear to me why the authors cannot integrate an existing heteroscedastic GP model (see below) with an existing batch BO algorithm and modify them for quantile optimization. Can the authors discuss this?

Regression with input-dependent noise: A Gaussian process treatment. NIPS 1997.

Most likely heteroscedastic Gaussian process regression. ICML 2007.

Variational heteroscedastic Gaussian process regression. ICML 2011.

A Distributed Variational Inference Framework for Unifying Parallel Sparse Gaussian Process Regression Models. ICML 2016.



It is not clear to me whether or how this work can be extended to the case of observation of uncontrolled variables which is common in related works on quantile optimization like (Cakmak et al., 2020) and the missing references below:

Distributionally robust Bayesian quadrature optimization. AISTATS 2020.

Optimizing conditional value-at-risk of black-box functions. NeurIPS 2021.

Value-at-risk optimization with Gaussian processes. ICML 2021.

Will the authors be able to find a common setting (perhaps at the expense of some unused features in the tested algorithms) for empirical comparison? Can the authors discuss the above?



Page 2: The authors say that "using a dataset... that does not necessarily require replicates of observations at the same input location." The dataset contains quantile values. Can the authors explain how these quantile values are obtained without replicating observations using the information from Section 2.1?

Page 2: From page 1, g can denote a quantile. Can the authors clarify whether y_1 to y_n are therefore quantile values?

Page 2: Like what the authors have said "While the literature on expectile regression is less extended", can the authors then provide a discussion to motivate the use of equation 3 and expectile regression and why we prefer it over quantile regression?

Page 2: The authors say that "the Bayesian expectile model we just introduced is new to the best of our knowledge." Since it is new, can the authors provide further explanation and justification for the form of equation 6?

Page 3: The authors say that "In practice, we choose a Gaussian prior for g and a log-Gaussian prior for sigma". Can the authors explain and justify these choices of GP and log-GP for the quantile/expectile g and the variance of the asymmetric Laplace/Gaussian distribution of epsilon, respectively?

Page 3: The authors say that "The variational parameters can be optimised jointly with the model parameters (e.g. mean function coefficients or kernel hyperparameters) such that Kullback-Leibler divergence between the approximate and the true posterior is as small as possible." This claim may not be correct if the model parameters and Z are not represented in a form based on *true* variational parameters. See, for example, the justification given in paragraph of the following reference:

Stochastic Variational Inference for Bayesian Sparse Gaussian Process Regression. IJCNN 2019.

Page 4: The authors say that "one may achieve the same result by sampling a trajectory from the posterior of g". Can the authors define the trajectory and explain how it is sampled?

Page 4: The authors say that "The main drawback of GP-based TS is the cost of sampling a trajectory, which can only be done exactly at a finite number of input locations at a cubic cost in the number of locations." Can the authors explain in detail why the cost remains cubic when utilizing the sparse variational GP model in Section 2.2?

Page 4: The authors say that "The procedure for generating quantile samples from the variational posterior of g can be summarised as follow: ..." I find the summary too vague for understanding. Can the authors provide the exact technical details of this procedure, provide justification for each step of this procedure, and finally give the pseudocode for this procedure? For example, when the authors compute the mean function m(x) of a GPR model, is this the posterior mean? s(.) is also not defined. How is this procedure exactly tied to the inference procedure in Section 2.2? How exactly does this procedure account for batch sampling?

Page 5: The authors say that "we no longer have expressions for the first term of (11) — the joint differential entropy of B-dimensional asymmetric Laplace variables". The first term is not conditioned on g. Can the authors explain why then are they B-dimensional asymmetric Laplace variables?

Page 5: The authors say that "However, the Q-GIBBON formulation makes it particularly well-suited for a greedy approach, where we first optimise Q-GIBBON for B = 1, then optimise for B = 2 while fixing the first point to the previously found value, etc. until B points are found." Can the authors provide the exact technical details for this approach? For example, for subsequent points being sampling (i.e., B>=2), does it depend on the quantile value of the first point?

Section 4.4: Can the authors verify that the Lunar Lander task indeed requires the modeling of heteroscedasticity? Why 10% quantile of reward and not others?

The only method of comparison is naive and not competitive empirically. Can the authors instead compare with existing competitive batch BO algorithms with a smaller number of replications per point in the batch, considering that the batch size is relatively large in some experiments? Is it necessary to utilize such a large number of replications per point in these experiments?



Minor issues

Figure 1: heteroscedasticy

Abstract and Section 3: Max-value entropy search is not the same as entropy search. The authors should revise the latter to the former.

Page 5: conjugancy

Page 5: leading to our propose

**Q7 Justification For Your Score:**

The cons (Q4, especially the main issue) outweigh the pros (Q3).

**Q9 Complying With Reviewing Instructions:**

1: Yes.

---

### Decision · Program_Chairs · 2022-05-15

**Decision:**

Accept (Poster)

**Comment:**

Meta Review: The paper proposes a method for Bayesian quantile and expectile optimization which changes the focus in BO from a mean estimate to a quantile of the estimated objective function. The goal is to provide a robust method with respect to extreme events. The proposed approach handles heteroskedasticity by using two latent GPs and large datasets with sparse inducing points approximations. Finally the authors propose two adaptations of classical acquisition functions to the quantile case.

The authors have replied to the reviewer's rebuttal.

Recommendation to the authors: please carefully review the paper as suggested by the Reviewers, also in order to improve the quality of the presentation.